# Monooxygenases and Antibiotic Resistance: A Focus on Carbapenems

**DOI:** 10.3390/biology12101316

**Published:** 2023-10-09

**Authors:** Daniela Minerdi, Davide Loqui, Paolo Sabbatini

**Affiliations:** 1Department of Agricultural, Forestry and Food Sciences, University of Turin, Largo Paolo Braccini 2, 10095 Grugliasco, TO, Italy; paolo.sabbatini@unito.it; 2Emergency Department, Città della Salute e della Scienza of Turin, 10100 Turin, TO, Italy; dloqui@cittadellasalute.to.it

**Keywords:** antibiotic resistance, carbapenem resistance, flavin monooxygenases, Baeyer–Villiger monooxygenases

## Abstract

**Simple Summary:**

Antibiotics are medicines used to prevent and treat infections in humans, animals, and plants. Antibiotic resistance is a naturally occurring phenomenon that has emerged as one of the most significant threats to global health and food security. Bacteria have the ability to acquire mutations that render them resistant to antibiotic molecules, contributing to the spread of resistance. The misuse and overuse of antibiotics have played a pivotal role in driving the development and proliferation of antibiotic resistance. The United Nations is estimating that by 2050, up to 10 million human deaths each year will be caused by the “superbugs”, very dangerous pathogens resistant to multiple antibiotic molecules. There will be health and macroeconomic consequences for the world if antimicrobial resistance is not tackled. The availability of antibiotics without prescription contributes to their overuse and misuse. Urgent measures are required to mitigate these issues and their substantial global impact. One of these measures is the search for new antibiotics designed on the basis of new targets of resistance. In this review, we show that enzymes called flavin monooxygenases could be a new and so-far-underseen candidate.

**Abstract:**

Carbapenems are a group of broad-spectrum beta-lactam antibiotics that in many cases are the last effective defense against infections caused by multidrug-resistant bacteria, such as some strains of *Klebsiella pneumoniae*, *Escherichia coli*, *Pseudomonas aeruginosa,* and *Acinetobacter baumannii.* Resistance to carbapenems has emerged and is beginning to spread, becoming an ongoing public-health problem of global dimensions, causing serious outbreaks, and dramatically limiting treatment options. This paper reviews the role of flavin monooxygenases in antibiotic resistance, with a specific focus on carbapenem resistance and the recently discovered mechanism mediated by Baeyer–Villiger monooxygenases. Flavin monooxygenases are enzymes involved in the metabolism and detoxification of compounds, including antibiotics. Understanding their role in antibiotic resistance is crucial. Carbapenems are powerful antibiotics used to treat severe infections caused by multidrug-resistant bacteria. However, the rise of carbapenem-resistant strains poses a significant challenge. This paper explores the mechanisms by which flavin monooxygenases confer resistance to carbapenems, examining molecular pathways and genetic factors. Additionally, this paper highlights the discovery of Baeyer–Villiger monooxygenases’ involvement in antibiotic resistance. These enzymes catalyze the insertion of oxygen atoms into specific chemical bonds. Recent studies have revealed their unexpected role in promoting carbapenem resistance. Through a comprehensive analysis of the literature, this paper contributes to the understanding of the interplay between flavin monooxygenases, carbapenem resistance, and Baeyer–Villiger monooxygenases. By exploring these mechanisms, it aims to inform the development of strategies to combat antibiotic resistance, a critical global health concern.

## 1. Introduction

Antibiotics, discovered approximately a century ago, are often regarded as miraculous drugs that have revolutionized the treatment of infectious diseases. They have been a cornerstone of modern medicine, and their availability has significantly contributed to public health improvements. However, their effectiveness and widespread use have sometimes been taken for granted. Lately, the efficacy of administered antibiotics has changed dramatically, giving rise to the phenomenon of antibiotic resistance, through which more and more human pathogenic bacteria have acquired the capability to withstand or tolerate the effects of an attack by one or multiple antibacterial molecules. Conventional treatments are becoming ineffective, and the loss of effectiveness in even last-resort antibiotics has become a grave concern in recent years [1]. This development has had significant consequences, including the persistence of infections and an increased risk of their spread to others. Every year, 700,000 people die globally from antibiotic resistance [2] and, by 2050, the United Nations estimates that the superbugs and associated forms of multidrug resistance are projected to cause the deaths of up to 10 million people annually, which is comparable to the number of deaths caused by cancer. Additionally, it is estimated that this global health threat will result in a staggering economic cost of approximately USD 100 trillion [2]. Bacteria causing common blood-stream infections, pneumonia, urinary tract infections are developing antibiotic resistance all over the world. A high percentage of nosocomial infections are caused by multidrug-resistant bacteria (i.e., Gram-negative bacteria resistant to all β-lactam antibiotics, methicillin-resistant *Staphylococcus aureus*, vancomycin-resistant enterococci). Patients affected by these types of infections not only have a high risk of death but also consume a substantial amount of healthcare resources, leading to significant health and macroeconomic consequences, particularly for emerging economies. The reliance on antibiotics for the treatment of infections places a strain on healthcare systems, resulting in increased healthcare costs and resource allocation challenges. In emerging countries, the economic burden of antibiotic use is particularly profound. Limited healthcare budgets and infrastructure strain are exacerbated by the high demand for antibiotics due to infectious disease prevalence. The cost of acquiring antibiotics, administering treatments, and managing potential complications adds to the economic burden, diverting resources that could be allocated to other critical healthcare needs. About 100 years after the first antibiotic treatment was given to the first infected patient, bacterial infections such as gonorrhea, pneumonia, and tuberculosis have become a threat once again, making the world face the possibility of a post-antibiotic era in which common infections could kill again unless immediate and counteractions are taken. About 70% of all pathogenic bacteria are resistant to at least one commercially available antibiotic [3] and antibiotic resistance may have worsened due to COVID-19 because of the overuse of antibiotics in humans [3].

Effluent from hospitals, agricultural runoff, and wastewater treatment facilities pose potential pathways for the dissemination of antibiotic-resistant bacteria and their associated resistance genes within soil and the surrounding ecosystems. Factors contributing to antibiotic resistance include the excessive and unregulated use of antibiotics, inadequate treatment processes, and the recycling of wastewater. This issue is further exacerbated by the misuse of antimicrobials within the medical, agricultural, and veterinary sectors. Additionally, the widespread presence of antimicrobial resistance within the food supply chain can be attributed to the excessive or inappropriate use of antimicrobials to combat infections in animals, plants, and humans. Moreover, the routine administration of antibiotics for growth promotion and disease prevention in healthy animals is eroding the effectiveness of these medications. Antimicrobial resistance is pervasive at the animal–human–environment interface, with the farm-to-fork continuum emerging as a potential reservoir for resistance development and dissemination. These environments may serve as repositories for genes associated with antibiotic resistance, which could potentially be acquired by pathogens, beneficial microorganisms related to food production, or other entities. Recognizing the food chain as a significant source of antimicrobial-resistant microorganisms highlights the potential for their dispersion at various stages of food production [4]. Until antibiotics had been synthetized, bacterial antibiotic resistance had been a natural phenomenon of adaptation of bacteria to toxic molecules in the environment for more than 2 billion years.

The anthropogenic application of a great amount of antibiotics in medicine and agriculture for many years has led to a significant evolution of the mechanism of resistance in a relatively short period of time.

In 2018, a research investigation examined the public’s awareness of antibiotic prescription, antibiotic resistance, and antibiotic-related topics within the United States. The findings underscored the pivotal role of public health initiatives in the ongoing battle against antibiotic resistance. Furthermore, the study recommended that forthcoming campaigns should focus on enhancing the public’s comprehension of antibiotic resistance while addressing behaviors that can be modified to mitigate the development of resistance [5].

## 2. Antibiotic Resistance Mechanisms

The ability of bacteria to withstand the effects of antibiotic molecules can be attributed to several factors, some of which are inherent to the bacteria themselves. This intrinsic resistance arises from specific characteristics within the bacterial structure or physiology. Specifically, the ability to withstand the effects of antibiotic agents may originate from inherent bacterial mechanisms. These mechanisms could result from the absence of a specific target, variations in the composition of the cytoplasmic membrane, or the antibiotic molecule’s incapability to traverse the outer membrane. In parallel with intrinsic resistance, bacteria possess the capability to acquire resistance through diverse mechanisms, which usually encompass a minimum of four distinct methods. These methods not only underscore the adaptability of bacterial species but also emphasize the multifaceted nature of antibiotic resistance development.

### 2.1. Modification of Antibiotic Target

Modification of the target sites of antibiotics represents a frequently observed mechanism of antibiotic resistance. Clinical strains displaying resistance can be identified across all classes of antibiotics, irrespective of their mechanisms of action. This resistance phenomenon arises from a series of genetic mutations that can affect various components, including the gene responsible for encoding the target protein, proteins involved in drug transport, and those associated with drug activation when exposed to the antibiotic [6].

#### 2.1.1. Decreased Permeability

Small hydrophilic antibiotics, such as β-lactams, utilize porins (Omp), which are proteins forming aqueous channels in the outer membrane of Gram-negative bacteria, to facilitate their entry into bacterial cells. Resistance to β-lactam drugs often stems from the antibiotic molecules’ inability to reach their intended targets, primarily due to a reduction in their passage through the porin channels in the bacterial outer cell walls [7]. It is worth noting that the number and structure of porins can vary among different Gram-negative bacterial species. For instance, *Escherichia coli* and *P. aeruginosa* exhibit intrinsic resistance to a wide array of antibiotics due to their reduced expression of high permeability porins [8]. This variation in porin composition contributes to the differences in antibiotic susceptibility seen across various bacterial strains. This revision provides a clearer explanation of how porins function and how changes in porin expression can lead to antibiotic resistance in specific bacterial species.

#### 2.1.2. Overexpression of Drug Efflux Pumps

In situations where intact antimicrobial agents enter bacterial cells, and the drug targets are readily available, active drug efflux systems frequently become involved. Bacteria efflux proteins are proteins present in both Gram-negative and Gram-positive bacteria, and they are divided into seven families. Drug resistance often arises in conjunction with the presence of drug efflux pumps, which function to reduce the accumulation of drugs within bacterial cells. Multidrug efflux pumps possess the capability to recognize various antibiotic molecules across different classes of action. In many cases, drugs characterized by diverse molecular structures, upon entering the bacterial cell through the periplasmic space or cell membrane, fail to reach their intended target sites due to active exportation by these efflux pumps. Clinical analyses of multidrug-resistant strains have consistently revealed an increased expression of genes responsible for encoding multidrug efflux pumps [9]. These genes are typically found on plasmids and other mobile genetic elements [10], facilitating their transfer between bacterial cells and contributing to the widespread dissemination of drug resistance.

#### 2.1.3. Hydrolysis of Antibiotic Molecules

The modification of antibiotics via hydrolysis represents a significant mechanism of antibiotic resistance. β-lactam-hydrolyzing enzymes have expanded their range of activity from penicillinases to cephalosporinases and have subsequently extended to spectrum β-lactamases (ESBLs), mannose-binding lectins (MBLs), and other carbapenemases [11]. The rise in ESBL-producing bacteria has not only increased the clinical usage of carbapenems but has also elevated carbapenem-hydrolyzing activity. These enzymes, referred to as carbapenemases, exhibit considerable diversity. They were initially identified in *Enterobacteriaceae*, categorized into the Ambler four classes of β-lactamases (class A, B, C, and D), and all share a serine component in their target region. The primary function of carbapenemases is to deactivate carbapenem antibiotics. While initially discovered on the chromosomes of specific bacterial species, it is noteworthy that many carbapenemases are plasmid mediated. As a result, they have been identified in various bacterial groups, including *Enterobacteriaceae*, *Pseudomonas aeruginosa*, and *Acinetobacter baumannii* [11].

Horizontal gene transfer (HGT) plays a crucial role in facilitating the rapid dissemination of antibiotic resistance, although the dynamics governing the transmission of genes responsible for conferring antibiotic resistance remain inadequately understood. HGT encompasses several mechanisms that deviate from the traditional vertical inheritance of genes. These mechanisms include conjugation mediated by plasmids, transduction facilitated by bacteriophages, and natural transformation through the uptake of extracellular DNA. Collectively, these processes enable the movement of genetic material across different strains and even between distinct species. Consequently, HGT introduces a pivotal dimension to the dynamics of infectious diseases. In this context, an antibiotic resistance gene (ARG) can serve as a pivotal factor in disease outbreaks by effectively transferring resistance traits to multiple unrelated pathogens. This phenomenon underscores the interconnectedness of microbial populations and highlights how the exchange of ARGs can amplify the challenge of combatting antibiotic-resistant infections. Surprisingly, one of the unintended consequences of antibiotic use in bacterial populations is the stimulation of their genetic evolution and the promotion of horizontal gene transfer, leading to increased antibiotic resistance dissemination [12]. This phenomenon is particularly evident within the gut microbiota, which serves as a rich reservoir of mobile antibiotic resistance genes. Notably, sub-inhibitory concentrations of orally administered antibiotics can exert substantial selective pressure in this environment, as highlighted by [12]. Specifically, when *E. coli* donor strains are exposed to sub-inhibitory concentrations of orally administered antibiotics like fluoroquinolones, such as ciprofloxacin and levofloxacin, the rates of plasmid conjugation for several plasmid types encoding broad-spectrum β-lactamases increase. This heightened conjugation frequency appears to be associated with antibiotics that inhibit DNA synthesis and is likely linked to the generation of reactive oxygen species (ROS) and potentially an upregulation of genes involved in the SOS response. Interestingly, certain antioxidant molecules, including edaravone, p-coumaric acid, and N-acetylcysteine, exhibit the ability to counteract the antibiotics’ capacity to enhance plasmid conjugation rates. This suggests that a combination of these antioxidative agents with inducer antibiotics might be a potential strategy to mitigate the increased dissemination of resistance plasmids, as proposed by Ortiz de la Rosa [12].

### 2.2. Antibiotic Resistance Mediated by Bacterial Enzymes

Bacteria during evolution have adopted several mechanisms of resistance to antibiotics; in some, the cell uses its own genes to survive antibiotic exposure; in others, the bacteria are able to survive thanks to new capacities attained by the acquisition of new genetic material that allow survival. Excellent reviews deal with this subject [13]. The aim of this review is to focus on the mechanisms of resistance mediated by bacterial proteins, specifically on the enzymes that modify drugs in detail. This paper aims to specifically highlight the role of monooxygenases in antibiotic resistance, both in a general context and with a specific focus on the involvement of Baeyer–Villiger monooxygenases in carbapenem resistance. The role of bacterial proteins in the development of antibiotic resistance is versatile and it includes (i) drug alteration (a) enzymes modify the antibacterial drug by destroying its antibacterial activity; (ii) drug alteration (b) that can occur through an enzymatic process where an enzyme covalently transfers various chemical groups to the drug, thereby preventing its binding to its intended target; (iii) target protection (a) proteins bind to the target of the antibiotic molecules, leading to allosteric dissociation of the drug from its intended target; (iii) target protection (b) proteins can bind to the antibiotic target causing a conformational change that allows the functioning of the target protein even in the presence of the drug; and (iv) target bypass where the enzyme target of the antibiotic molecule becomes redundant thanks to the acquisition of a gene that encodes an alternative enzyme that fulfils the function of the drug target [14] (Figure 1).

Oxidoreductases, transferases, and hydrolases (Figure 1) are the main classes of enzyme responsible for the antibiotic resistance. Among hydrolases, the most common enzymes that catalyze antibiotic hydrolysis are β-actamases and macrolide esterases that destroy β-lactams, macrolides, chloramphenicol, and phosphomycin. β-lactamases hydrolyze the amide bond in the β-lactam ring, the common structural element of all β-lactam antibiotics like penicillins, cephalosporins, carbapenems, and monobactams [15]. The emergence of high rates of new mutations in the genes coding for β-lactamases that produce enzymes with a different structure and their location on mobile genetic elements contributes to the rapid spread of resistant bacteria [16]. Macrolide esterases are implicated in macrolide detoxification. Macrolides are a class of antibiotics extensively used in both agriculture and medicine. Erythromycin and azithromycin have been used as substitutes for β-lactam antibiotics in patients with penicillin allergies [17]. Transferases are a class of enzymes with a different substrate specificity, type of modification, and mechanism of action that modify the antibiotic molecules by covalently binding different chemical groups [17].

#### 2.2.1. Flavin-Dependent Monooxygenases (FMOs)

Flavin-dependent monooxygenases (FMOs) insert one atom of oxygen into an organic substrate, reducing the other oxygen atom to water. They use oxygen and a reductant, such as NADPH, to oxygenate many compounds [18]. FMOs catalyze redox reactions involved in the detoxification of drugs, in the biosynthesis of hormones and antibiotics, in the catabolism of xenobiotic compounds, and in the hydroxylation of amino acids [19]. The catalytic cycle of FMOs comprises the reductive and oxidative reactions (Figure 2). An equivalent form of NAD(P)H is used to reduce the flavin cofactor and to activate molecular oxygen. Oxygen activation transfers a single electron to form a flavin-semiquinone and a superoxide radical pair that combine to form C(4a)-hydroperoxyflavin (Fl_4a_-OOH), which oxidizes the substrate [20]. Fl_4a_-OOH catalyzes the epoxidation, hydroxylation, and Baeyer–Villiger oxidation of a wide range of substrates [21] (Figure 2).

FMOs are classified into eight classes based on amino acid sequence and biochemical and structural properties. The eight classes (A-H) are divided into two groups based on mechanistic properties. Subclasses A, B, G, and H are single-component monooxygenases, and they are reduced and oxidized by an electron donor within the same polypeptide chain. The subclasses C–F are two-component monooxygenases that obtain the reduced flavin from a flavin oxidase enzyme [22].

#### 2.2.2. Baeyer–Villiger Monooxygenases (BVMOs)

Baeyer–Villiger monooxygenases (BVMOs) are enzymes containing flavin as cofactor and oxidize the carbonyl group of the substrate performing the so called Baeyer-Villiger reaction. Microbial BVMOs convert ketones into the corresponding esters and lactones, giving carbon units available for further catabolism. BVMO enzymes are present in bacteria, fungi, and plants but they are absent in humans and animals. BVMOs possess a flavin cofactor and, as an electron donor, they use the NADH or NADPH required. 

BVMOs are a class of soluble enzymes that can be categorized into three distinct types [22]. Enzymes belonging to type I are characterized by the presence of an FAD cofactor, a dependency on NADPH, and the possession of two Rossmann folds to bind dinucleotide. They also feature the Baeyer–Villiger motif [FXGXXXHXXXW(P/D)] crucial for their catalytic activity [23]. In contrast, enzymes of class II utilize flavin mononucleotide and NADH as a co-substrate. Lastly, type 0 BVMOs do not have the BVMO motif and employ flavin adenine dinucleotide (FAD) and NAD(P)H [24]. BVMOs have significant medical implications, exemplified by enzymes like ethionamide monooxygenase (EtaA), which is a type I BVMO. EtaA has the capacity to convert various ketones into their respective esters [25]. Additionally, this enzyme can oxidize the sulfide components of several antitubercular thioamide drugs [26]. The oxidized drugs exhibit high toxicity to mycobacteria, underscoring the role of ethionamide monooxygenase as a prodrug activator. Furthermore, there is a heme-containing BVMO that belongs to the cytochrome P450 superfamily [27]. 

### 2.3. FMOs and Antibiotic Resistance

#### 2.3.1. Tetracyclines

Microbial FMOs are able to modify antibiotics through oxidation, leading to resistance. The class A flavin-dependent monooxygenase tetracycline destructase TetX confers resistance to all clinically relevant tetracyclines, including the broad-spectrum antibiotic tigecycline, which is successfully used against multidrug-resistant pathogens. TetX was isolated from transposons *Tn*4351 and *Tn*4400 present in anaerobic bacteria of the genus Bacteroides [28]. It is able to regioselectively hydroxylate tetracycline antibiotics to 11a-hydroxy-tetracyclines (Figure 3A). Overexpression in *E. coli* and protein purification followed by mass spectral and NMR characterization demonstrated that TetX requires NADPH, Mg^2+^, and molecular oxygen to hydroxylate tetracyclines at C11a. This hydroxylation weakens the binding to magnesium, altering the physical properties of tigecycline and decreasing its affinity to ribosomes. Intramolecular cyclization and non-enzymatic breakdown to non-defined products follow the hydroxylation step [29]. The crystallographic structure of TetX from *Bacteroides thetaiotaomicron* and its complexes with tetracyclines showed the extremely versatile substrate diversity of the enzyme [30]. The environmental bacterium *Mycobacterium abscessus* has emerged as an important human pathogen causing bronchopulmonary infections. It possesses a WhiB7-independent tetracycline-inactivating monooxygenase, MabTetX, that confers a high level of resistance to tetracycline and doxycycline. Both antibiotic molecules are monooxygenated by the purified MabTetX [31]. 

Tet (56), which shares similarity with TetX, has been identified in *Legionella longbeachae*, the pathogen responsible for causing Legionnaires’ disease. Detailed analysis using X-ray crystal structure revealed that Tet 56 possesses a structural configuration comprising a flavin adenine dinucleotide (FAD)-binding Rossmann-type fold domain, a domain responsible for binding tetracycline, and a C-terminal α-helix that serves as a connecting link between these two domains [32]. Additionally, genes encoding TetX3 and TetX4 have been detected in a multitude of *Enterobacteriaceae* and Acinetobacter strains originating from both animals and humans [32].

#### 2.3.2. Rifamycins

The rifamycins represent a group of naturally occurring compounds, as well as their semi-synthetic derivatives, characterized by their activity against a wide spectrum of bacteria, encompassing both Gram-positive and Gram-negative species. These compounds possess a structural framework that incorporates a naphthalene core, connected via a polyketide “handle” and linked to the naphthalene segment through a cyclic amide bond. Their three-dimensional configuration resembles that of a basket, a crucial feature enabling them to bind effectively to the RNA exit tunnel located on the bacterial RNA polymerase’s β subunit, which serves as their target site [33]. The enzyme rifampicin monooxygenase (RIFMO) is another class A FMO that modifies rifampicin (RIF), an antibiotic that acts by inhibiting DNA-dependent RNA polymerase, used in combination therapy for the treatment of tuberculosis and mycobacterial and non-mycobacterial infections [34]. *Nocardia farcinica* [35] as well as *Streptomyces* and *Rhodococcus* species present in the environment possess the *rifmo* gene [35]. RIF was proposed to be hydroxylated by RIFMO to produce 2′-N-hydroxy-4-oxo-RIF, leading to subsequent RIF decomposition [36] (Figure 3B), but it was unclear the mechanism by which RIF was modified and degraded. Years later, it was described in *Streptomyces venezuelae*, with the monooxygenation of position 2 of the naphthyl group with a consequent ring opening and linearization of the antibiotic, leading to an antibiotic that cannot adopt the basket-like structure that is essential for binding to the RNA exit tunnel of the target ribosome site [37]. Liu and colleagues [38] successfully elucidated the crystal structure of RIFMO in association with the hydroxylated product of RIF. Their structural analysis revealed a notable disruption in the ansa aliphatic chain of RIF, positioned precisely between the naphthoquinone C2 and amide N1. This observation strongly implies that RIFMO hydroxylates RIF at the C2 atom, subsequently leading to the cleavage of the ansa linkage. This cleavage event ultimately results in the deactivation of the antibiotic properties of RIF by preventing critical interactions with its target, the RNA polymerase.

#### 2.3.3. Sulfonamides

Sulfonamides were synthesized and introduced into the environment approximately 90 years ago. This relatively short timeframe has limited the opportunity for bacteria to evolve resistance to these compounds [39]. Nevertheless, sulfonamide-degrading bacteria have been found in seawater, agricultural soil, activated sludge, and acclimated membrane reactors, suggesting the existence of resistance mechanisms.

The bacterial breakdown of sulfonamides has dual significance in environmental cleanup by eliminating pollutants and in the context of antibiotic resistance, where the enzymes responsible for degradation can be viewed as potential mechanisms for resistance development. SadA and SadB are two monooxygenases using the FMN reductase SadC that enable *Microbacterium* sp. strain BR1 and other Actinomycetes to inactivate sulfonamide antibiotics, using them as a carbon source. SadA and SadC attack the sulfonamide molecules and release 4-aminophenol, which is converted by SadB to 1,2,4-trihydroxybenzene that lacks antibiotic activity [40] (Figure 3C).

*Microbacterium* sp. CJ77 possesses a two-component class D FMO system encoded by the monooxygenase gene *sul*X and the flavin reductase gene *sul*R [41] and are similar to the *Microbacterium* sp. strain BR1 SadA/SadB system. The presence of sulfonamides upregulates the expression of the *sul*X and *sul*R genes. Sulfonamide monooxygenase and flavin reductase were expressed and purified in *E. coli*, and biochemical analysis showed that sulfamethazine is cleaved by the two-component monooxygenase system to produce 4-aminophenol and the dead-end products (Figure 3D) [41]. *Sul*R and *sul*X genes were co-expressed in *E. coli*, conferring decreased susceptibility to sulfamethoxazole, suggesting that the two genes are potential resistance determinants and encode drug-inactivating enzymes. All sulfonamide-degrading actinobacteria possess the gene cluster *sul*X-*sul*R in a genomic island also carrying class 1 integrons. This implies that the ability to metabolize sulfonamides may have been obtained by sulfonamide-resistant bacteria that had previously acquired the class 1 integron due to selective pressures from sulfonamide exposure [41].

## 3. Carbapenems

Carbapenems belong to a class of semi-synthetic β-lactam antibiotics derived from thienamycin, a natural compound produced by the soil microorganism Streptomyces cattleya, which was discovered in 1970 [42]. They are characterized by their wide spectrum of activity and exceptional potency against both Gram-negative and Gram-positive bacteria. Consequently, they are reserved as a last-line treatment for patients suffering from severe infections or suspected of having multidrug-resistant bacterial infections [43]. Carbapenems exhibit relative resistance to hydrolysis by most β-lactamases. Their structural composition consists of a β-lactam ring, which is similar to penicillins (“penams”) fused with an unsaturated five-membered ring. The key structural difference lies in the presence of a double bond between C-2 and C-3 (“-penem”) and the presence of carbon (“carba-”) at position 1 [44]. The presence of this carbon is fundamental for their spectrum of activity, stability against β-lactamases, and potency. They resist most β-lactamases due to the presence of a trans hydroxyethyl side chain, which replaces the acylamino substituent found on the β-lactam ring in penicillins and cephalosporins. Although carbapenems acylate the serine residue on β-lactamases rapidly, the subsequent hydrolysis of the acylated enzyme is very slow because the trans-1-hydroxyethyl moiety displaces the water necessary for hydrolysis at the active site. As a result, the enzyme becomes acylated and loses its activity [45]. Carbapenems are hence considered highly reliable as last-resort drugs for treating bacterial infections. Carbapenems exhibit varying antibacterial activities. Some, like panipenem, imipenem, and doripenem, are effective against Gram-positive bacteria, while others, including ertapenem, biapenem, meropenem, and doripenem, have limited effectiveness against Gram-negative bacteria [46]. They are preferred over other antimicrobial agents for treating invasive or life-threatening infections due to their concentration-independent bactericidal effect and minimal adverse effects [47]. Carbapenems like doripenem, tebipenem, meropenem, imipenem, ertapenem, biapenem, and panipenem are used worldwide, especially in response to the increasing resistance to cephalosporin antibiotics within the *Enterobacteriaceae* group.

### 3.1. Mechanism of Action

Carbapenems cannot easily diffuse through the bacterial cell wall and, for this reason, they enter Gram-negative bacteria through porins’ outer membrane [48] and exhibit bactericidal activity by binding to Penicillin-Binding Protein (PBP) 1a, 1b, 2, and 3 [49]. 

PBPs (penicillin-binding proteins) are integral cytoplasmic membrane proteins responsible for constructing the peptidoglycan layer within the bacterial cell wall. They play a crucial role in maintaining the three-dimensional structure of the bacterial cell wall. Carbapenems, due to their structural similarity to acylated D-alanyl-D-alanine, have the unique ability to bind irreversibly to the active site of PBPs. When β-lactam molecules, such as carbapenems, bind to PBPs, they disrupt essential processes within the bacterial cell. Specifically, this binding inhibits the bacterium’s ability to carry out transpeptidation and other peptidase reactions within the peptidoglycan layer. As a result, the bacterium becomes incapable of properly forming its cell wall. This disruption in cell wall synthesis leads to the subsequent autolysis of the bacterial cell, ultimately causing its demise. This process is accompanied by the extrusion of the cell membrane through the weakened areas in the cell wall [49]. The hypertonic bacterial cell bursts via osmotic shock due to osmotic pressure because the membrane is too weak to prevent it. The affinity of carbapenems for specific PBPs depends on strains and species of bacteria. Carbapenems exhibit distinct characteristics based on their affinity for specific penicillin-binding proteins (PBPs). Notably, their affinity for PBP1a or PBP1b sets carbapenems apart from other β-lactam antibiotics. Additionally, their interaction with PBP-2 or PBP-3 in Gram-negative bacteria is a distinguishing feature that can differentiate one carbapenem from another [50].

### 3.2. Carbapenem Resistance in ESKAPE Pathogens

*Klebsiella pneumoniae*, *E. coli*, *P. aeruginosa*, and *A. baumannii* are among the highly concerning carbapenem-resistant bacteria that are part of the ESKAPE group. This group, which includes Enterococcus faecium, *Staphylococcus aureus*, *K. pneumoniae*, *A. baumannii*, *P. aeruginosa*, and Enterobacter species, collectively infect more than 2 million individuals in the United States annually. Tragically, their resistance contributes to approximately 23,000 deaths each year in the United States [51]. Antimicrobial-resistant ESKAPE pathogens represent a global threat to human health [52]. Among the *Enterobacteriaceae*, *E. coli* and *K. pneumoniae* constitute the greatest risk for public health because they are implicated both in community- and hospital-acquired infections. They exhibit a broad range of clinical symptoms, possess resistance to multiple drugs, and can swiftly transfer their resistance to other microorganisms [53,54]. Carbapenem-resistant K. pneumoniae, in particular, pose a significant challenge in clinical environments due to their extended antibiotic-resistant characteristics and their worldwide spread facilitated by mobile genetic elements [55]. Since the 1990s, hypervirulent strains of *K. pneumoniae* have emerged as superbugs, capable of causing pyogenic liver abscesses and other invasive syndromes. These strains have displayed various mechanisms associated with antibiotic resistance. It is worth mentioning that reduced susceptibility to carbapenem antibiotics has been associated with factors such as carbapenemase production, decreased expression, and the loss of outer membrane proteins OmpK35 and OmpK36 (referred to as OMPs), along with the overexpression of efflux pumps [56,57]. These factors collectively contribute to the challenges in treating infections caused by hypervirulent *K. pneumoniae* strains and highlight the importance of understanding their mechanisms of antibiotic resistance [58,59].

*E. coli* is a common bacterium found in the human gastrointestinal tract and is a natural part of the human bacterial flora. However, it is also a significant causative agent of conditions such as urinary tract infections, sepsis, and meningitis. Resistance to carbapenems in *E. coli* has been attributed to modifications in the non-specific outer membrane porins OmpC and OmpF, resulting in reduced membrane permeability [60]. Predictive analysis has suggested that amino acid substitutions in ompC may play a pivotal role in driving carbapenem resistance, while amino acid deletions could significantly impact ompF mutations [61,62]. Contrary to earlier indications, recent research has shown that antibiotic resistance in this context is not primarily linked to the overexpression of efflux pumps [61]. Instead, the main contributors to carbapenem resistance are the production of carbapenemases. These enzymes fall into different classes, including class A KPCs; class B metallo-β-lactamases like VIM, IMP, or NDM-1; and class D OXA-type enzymes such as OXA-48 [62]. These carbapenemases play a pivotal role in conferring resistance to carbapenem antibiotics, and understanding their presence and mechanisms is critical for effective treatment strategies.

*A. baumannii* is a significant nosocomial pathogen, particularly in intensive care units, where it is associated with critical healthcare-related diseases and infections. The global increase in carbapenem-resistant *A. baumannii* strains is a growing concern [63]. The mechanisms behind carbapenem resistance involve reduced drug affinity due to the downregulation of penicillin-binding proteins (PBPs) [64], as well as membrane impermeability caused by decreased expression or mutations in porins. Efflux pumps also play a role in carbapenem susceptibility in *A. baumannii* [65]. Among these pumps, AdeABC, AdeFGH, and AdeIJK are noteworthy, as they influence susceptibility to carbapenems in *Acinetobacter* species [66]. The most significant mechanism of carbapenem resistance in *A. baumannii* involves the inactivation or enzymatic degradation of carbapenems, typically catalyzed by carbapenemase enzymes. These enzymes are often located on plasmids and are highly transferable [67]. *P. aeruginosa* is frequently implicated in severe healthcare-associated complications, often manifesting as urinary tract, respiratory, and bloodstream infections. Notably, this bacterium exhibits intrinsic resistance to a range of antimicrobial agents [68]. This inherent multidrug resistance stems from a combination of factors, including the concerted action of broad-spectrum drug efflux pumps and a relatively low degree of permeability in the outer membrane. Specifically, carbapenem resistance in *P. aeruginosa* can be attributed to several mechanisms. These encompass the loss of the porin OprD, often due to the insertion of sequence elements. Additionally, the presence of multidrug efflux systems and the expression of chromosomal β-lactamases further contribute to the development of resistance to carbapenem antibiotics [68].

### 3.3. Incidence of Carbapenem Resistance

The increasing resistance to carbapenems among Gram-negative bacteria, particularly within the *Enterobacteriaceae* family, has become a major global concern. Carbapenem-Resistant *Enterobacteriaceae* (CRE) cases are on the rise worldwide, with significant occurrences documented in both the United States and Europe. This alarming trend has led to a surge in reported CRE-associated infections, especially those acquired in healthcare settings. Notably, the highest prevalence of CRE infections is observed in Mediterranean and Balkan countries, highlighting the pressing need to address this issue in these regions. Greece and Italy, for instance, report prevalence rates of approximately 60% and 40%, respectively [69,70]. In Asia, a study has shown that CRE infection accounts for 0.6% to 0.9% of all culture-positive infections [71]. While comprehensive data on Carbapenem-Resistant *Enterobacteriaceae* (CRE) in India may be limited, published articles provide valuable insights into the concerning situation. These sources indicate that the prevalence of carbapenem resistance among *Enterobacteriaceae* in India varies, ranging from 18% to 31%. This underscores the significant challenge posed by CRE in the Indian healthcare landscape, emphasizing the crucial need for comprehensive surveillance and intervention measures. 

Tesfa and their research team [72] conducted an extensive analysis encompassing 35 original studies focused on the colonization of carbapenem-resistant *K. pneumoniae* (CRKP). Their study also involved the examination of 32 records, encompassing data from 37,661 patients across 18 different countries. This comprehensive assessment spanned the years from 2010 to 2021 and aimed to determine the prevalence of CRKP colonization. Ten studies involving a total of 3643 patients admitted to the intensive care unit (ICU) were analyzed to assess the incidence of colonization. The data gleaned from these studies reveal a disturbing trend: colonization with carbapenem-resistant K. pneumoniae (CRKP) has been on the rise from 2010 to 2022. The prevalence of colonization with carbapenem-resistant K. pneumoniae varies significantly based on geographic location, with rates spanning from as low as 0.13% to as high as 22%. When considering all the data, a pooled prevalence of 5.43% emerges, indicating that a substantial portion of the population is carrying CRKP. This is deeply concerning, as colonization serves as a potential risk factor for infection. Notably, it is critical to recognize that CRKP infections come with a grave outcome, as referenced in Agyeman’s work [73], where it is revealed that one in every three individuals infected with CRKP succumbs to the infection. This underscores the urgency of addressing and mitigating the spread of CRKP colonization and infections within healthcare settings. The research findings indicate that the highest rates of colonization by carbapenem-resistant K. pneumoniae (CRKP) were observed in the Asian continent, particularly in countries such as China and India, where the frequency stood at 1.4%. In Europe, the frequency of CRKP colonization was slightly lower, at 1.2%, while in the Americas, it was 0.3%, and in Africa, the lowest frequency was recorded at 0.07%. It is important to note that the incidence of colonization varied significantly, ranging from 2% to as high as 73%, with an overall pooled incidence of 22.3%. Notably, most reports on both prevalence and incidence primarily originated from developed countries. CRKP was commonly detected on contaminated medical equipment, the hands of healthcare personnel, and within the gastrointestinal tracts of patients. Furthermore, the distribution of carbapenem resistance genes among the colonizing isolates exhibited noteworthy variations, with carbapenemase genes being a prominent finding across numerous studies. Additionally, New Delhi Metallo beta-lactamase genes were predominantly reported in studies conducted in Asian countries, as documented in the research by [74]. This divergence in gene distribution underscores the complex dynamics of carbapenem resistance and highlights the importance of region-specific insights in understanding and addressing this critical issue.

Hospital-acquired infections (HAIs) caused by *A. baumannii* (HA-AB), especially those involving carbapenem-resistant strains (HA-CRAB), present a significant and pressing healthcare concern, leading to therapeutic challenges, and endangering patients’ well-being on a global scale. The World Health Organization (WHO) has recently elevated carbapenem-resistant *A. baumannii* (CRAB) to a global priority pathogen. Ayobami and colleagues [75] conducted an extensive analysis of the incidence and prevalence of Hospital-Acquired *A. baumannii* (HA-AB) and Hospital-Acquired Carbapenem-Resistant *A. baumannii* (HA-CRAB) infections across regions categorized by the WHO, including Europe (EUR), Eastern Mediterranean (EMR), and Africa (AFR). Their study compiled data from 24 records, with 16 originating from Europe, 6 from the Eastern Mediterranean, and 2 from Africa, resulting in a comprehensive assessment drawing from 3340 records. The pooled estimates for the incidence and incidence density of HA-AB infections within intensive care units (ICUs) were determined to be 56.5 cases per 1000 patients (with a 95% confidence interval of 33.9–92.8) and 4.4 cases per 1000 patient-days (with a 95% confidence interval of 2.9–6.6), respectively. Regarding carbapenem-resistant *A. baumannii* (CRAB) infections in ICUs, the pooled incidence and incidence density were calculated at 41.7 cases per 1000 patients and 2.1 cases per 1000 patient-days. These findings highlight that within ICUs, *A. baumannii* and carbapenem-resistant *A. baumannii* strains constituted 20.9% and 13.6% of all hospital-acquired infections (HAIs), respectively. This underscores the continued clinical significance of hospital-acquired *A. baumannii* infections within the WHO-defined regions, especially within ICU settings. In another study conducted on fifty CRAB isolates collected from four major hospitals in Bahrain, it was observed that 98% of the isolates displayed resistance to imipenem and 98% to meropenem. A majority of these isolates carried resistance determinants such as blaO-XA-51, blaIMP, and blaOXA-23, confirming a high level of antibiotic resistance. Class D carbapenemases were prevalent in the examined isolate collection, and resistance genes were found in various combinations [76]. Carbapenem-resistant *P. aeruginosa* (CR-PA) is a significant healthcare-associated pathogen globally. In the United States, 10–30% of *P. aeruginosa* isolates exhibit carbapenem resistance, and this percentage varies across different regions worldwide. Carbapenemases present in *P. aeruginosa* show substantial regional variation and encompass class A beta-lactamases like KPC and GES, metallo-beta-lactamases such as IMP, NDM, SPM, and VIM, and the Class D OXA-48 enzymes [77]. Carbapenem resistance in *P. aeruginosa* exhibits significant global variation. In most countries, the prevalence of carbapenem resistance in this bacterium ranges from 10% to 50%. To provide context, some countries like Canada and the Dominican Republic have reported relatively low rates, with figures as low as 3.3% and 8%, respectively. In contrast, regions such as Australia, North America, and certain European countries report rates ranging from 10% to 30%. However, there are areas of particular concern for public health due to notably high carbapenem resistance rates. Countries such as Saudi Arabia, Iran, Poland, Greece, Russia, Costa Rica, Peru, and Brazil have reported rates exceeding 50%. These regions stand out as having a predominant prevalence of high resistance rates, indicating a significant challenge in managing and addressing this public health concern, as highlighted by Hong’s research. Regarding carbapenemases, a report by the Centers for Disease Control and Prevention in 2018 indicated that 1.9% of carbapenem-resistant *P. aeruginosa* isolates were carbapenemase producers [78]. Worldwide, the dissemination of carbapenemases varies significantly, with metallo-β-lactamases remaining the most predominant, especially the VIM group, followed by IMP and NDM. Others have shown a regional spread, such as SPM in Switzerland, China, Brazil, United Kingdom and India; DIM in Poland; SIM in China; AIM in Australia. HMB in Germany and the United States; GIM in Germany; CAM in Canada; and FIM in Italy. In Asia and Europe, researchers have identified *P. aeruginosa* clones that produce carbapenemases, particularly KPC and GES. These enzymes confer resistance to carbapenem antibiotics. However, the least identified carbapenemase type in these regions is the OXA-type, which has been reported in specific countries, including Spain, India, the United Kingdom, and Belgium, as detailed in Yoon’s research. This diversity in carbapenemase types highlights the complex landscape of antimicrobial resistance and the importance of regional surveillance and control efforts.

## 4. BVMO and β-Lactams Resistance

### 4.1. Carbapenems

Minerdi and colleagues [79] made a pioneering observation, shedding light on a novel aspect of carbapenem resistance. Their study revealed that carbapenemases are not the sole culprits in carbapenem inactivation; Baeyer–Villiger monooxygenases (BVMOs) also possess this capability through oxygenation. This study focused on an *Acinetobacter radioresistens* strain isolated from the environment. This particular strain was found to harbor a Baeyer–Villiger monooxygenase known as Ar-BVMO. Remarkably, Ar-BVMO exhibited a complete amino acid sequence identity with the ethionamide monooxygenase found in multidrug-resistant *A. baumannii.* The significance of this finding is underscored by the fact that carbapenem antibiotics are considered the last-resort treatment for multidrug-resistant (MDR) *A. baumannii* infections. Unfortunately, there has been a global surge in the prevalence of *A. baumannii* strains that have developed resistance to these vital antimicrobial drugs [80]. 

Carbapenem resistance in *A. baumannii* mainly arises from acquired carbapenem-hydrolyzing oxacillinases such as OXA-23 that are classified under the Ambler class D β-lactamases [80]. Interestingly, the source of the blaOXA-23 gene is *A. radioresistens*, a commensal bacterial species commonly found on the skin of hospitalized and healthy patients [81]. Furthermore, two kinase inhibitors, tozasertib (VX-680) and danusertib (PHA-739358), previously identified as potential targets in anticancer therapy, were demonstrated to be metabolized by Ar-BVMO. To assess whether the expression of Ar-BVMO in imipenem-sensitive *E. coli* BL21 cells could confer resistance to imipenem, a disk diffusion assay was performed following EUCAST guidelines. According to the EUCAST clinical breakpoint table for *Enterobacteriaceae*, imipenem zone diameter breakpoints for sensitivity and resistance are ≥22 mm and <16 mm, respectively. Notably, *E. coli* BL21 cells transformed with pT7-Ar-BVMO but not induced, as well as *E. coli* BL21 cells transformed with an empty expression plasmid and induced with IPTG, displayed inhibition zones with average diameters of 25.5 mm and 28.0 mm, respectively. Conversely, *E. coli* BL21, when expressed in Ar-BVMO, showed resistance to imipenem (zone diameter breakpoint, <16 mm). These findings offered preliminary evidence that imipenem-sensitive *E. coli* expressing Ar-BVMO may become resistant to the antibiotic molecule [82].

The functionality of the purified Ar-BVMO enzyme in the presence of imipenem was evaluated by monitoring the utilization of NADPH. The derived kinetic parameters, Km and Vmax strongly suggested that imipenem could indeed serve as a genuine substrate for Ar-BVMO. Furthermore, liquid chromatography–mass spectrometry analysis revealed a classical Baeyer–Villiger asymmetric oxygen insertion within the carbapenem ring of imipenem (as depicted in Figure 3E). This compelling evidence confirms that imipenem qualifies as a substrate for Ar-BVMO. Remarkably, this marks the first instance of an antibiotic inactivating BVMO enzyme that, in the course of its typical BV oxidation activity, concurrently employs an unprecedented mechanism for carbapenem resistance. It is hypothesized that genetic exchange may have occurred between *A. radioresistens* and *A. baumannii* within the human body [83]. This outcome may have facilitated the transfer of the BVMO gene to the latter species. 

### 4.2. β-Lactams

Methicillin and/or vancomycin-resistant *Staphylococcus aureus* is one of the most common opportunistic human pathogens. It causes skin infections but also sepsis and pneumonia. Hwang and colleagues [84] described the crystal structure of a BVMO originating from Methicillin-resistant *Staphylococcus aureus* (SAFMO). This enzyme could potentially contribute to vancomycin and/or methicillin resistance by breaking down these molecules. Molecular docking simulations showed that vancomycin and methicillin can be accommodated in the SAFMO active site. If confirmed, this hypothesis could lead to a novel drug target against *S. aureus*.

## 5. Conclusions and Future Directions

This manuscript underscores the critical challenge of discovering novel antibacterial molecules based on new antibacterial targets and alternative bacterial resistance strategies. To address the pressing issue of antibiotic resistance, it is imperative to explore innovative avenues in antibiotic development. One promising direction highlighted in this review is the potential targeting of bacterial monooxygenases as a new therapeutic approach. By focusing on these enzymes, which play pivotal roles in bacterial metabolism and detoxification processes, we can potentially disrupt essential pathways and impair bacterial survival. Understanding the mechanisms and functions of monooxygenases can provide valuable insights into their potential as antibacterial targets. By elucidating their roles in antibiotic resistance, we can identify vulnerabilities in bacterial defenses and exploit them to develop effective interventions. Furthermore, exploring the interplay between monooxygenases and antibiotic resistance, particularly the recently discovered involvement of Baeyer–Villiger monooxygenases in carbapenem resistance, opens up new avenues for research. Unraveling the underlying mechanisms by which these enzymes mediate resistance can pave the way for the development of novel therapeutic strategies that specifically target and overcome such resistance mechanisms. The discovery of new antibacterial targets, such as bacterial monooxygenases, offers a fresh perspective in the fight against antibiotic resistance. It presents an opportunity to develop innovative therapies that can circumvent existing resistance mechanisms and provide effective treatments for bacterial infections. In conclusion, by exploring the potential of monooxygenases as antibacterial targets, this review underscores the importance of continuing research efforts in this field. The identification and development of new antibacterial molecules based on these targets hold immense promise in overcoming antibiotic resistance and ensuring the efficacy of future treatments. It is imperative to further investigate the mechanisms and functions of monooxygenases to harness their potential in the development of next-generation antibiotics. A final but important component of future research directions is the exciting potential target that could derive from quorum sensing. Quorum sensing is a communication mechanism used by bacteria to coordinate collective behaviors, including the expression of virulence factors and the formation of biofilms. Disrupting quorum sensing pathways can impair bacterial pathogenicity and potentially render bacteria more susceptible to existing antibiotics. By targeting quorum sensing, we can potentially interfere with the ability of bacteria to coordinate their resistance mechanisms and evade conventional antibiotic treatments. Developing compounds that specifically modulate quorum sensing pathways holds great promise for combating antibiotic resistance and enhancing the effectiveness of existing antibiotics. Further investigation into the underlying mechanisms of quorum sensing and its intricate relationship with antibiotic resistance is warranted. This knowledge can pave the way for the design and development of innovative strategies that disrupt bacterial communication and restore the efficacy of antibiotics. Continued research efforts in this field will contribute to the advancement of antibacterial strategies and the preservation of effective treatments for infectious diseases.

## Figures and Tables

**Figure 1 biology-12-01316-f001:**
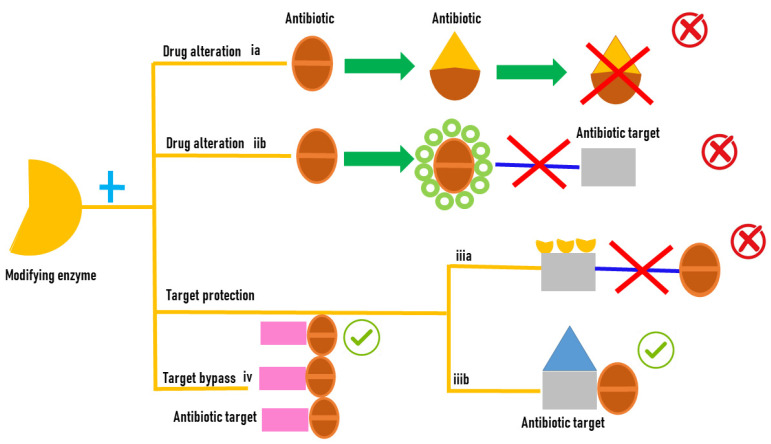
Antibiotic resistance mediated by enzyme modifications.

**Figure 2 biology-12-01316-f002:**
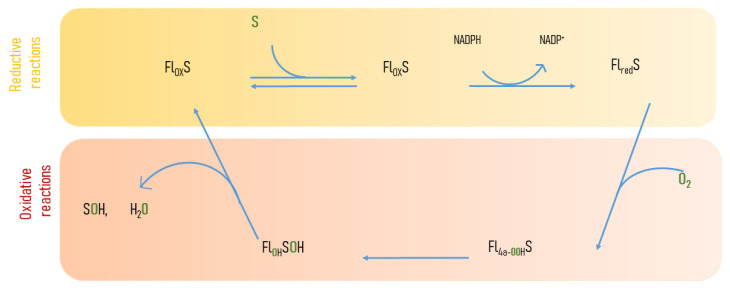
Catalytic cycle of flavin-containing monooxygenases. Reductive-half reaction is shown in yellow and the oxidative-half reaction in orange.

**Figure 3 biology-12-01316-f003:**
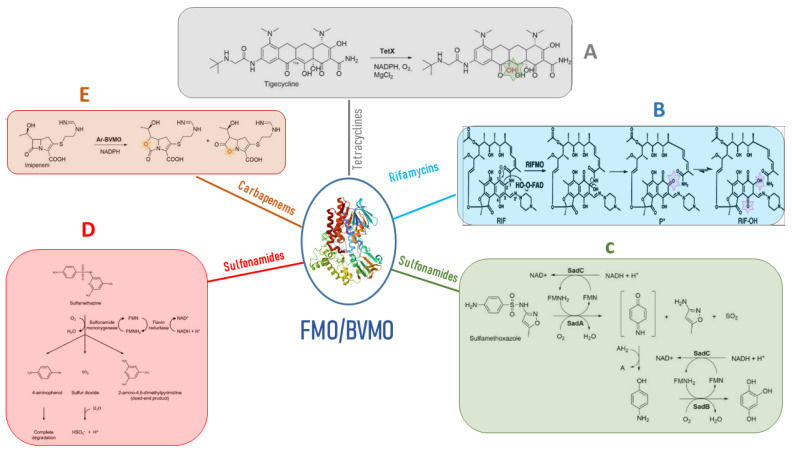
Antibiotic resistance mediated by FMOs’ modifications of antibiotic molecules. (**A**) the class A flavin-dependent monooxygenase tetracycline destructase TetX hydroxylates tetracycline antibiotics to 11a-hydroxy-tetracyclines inactivating the antibiotic molecule. (**B**) Rifampicin monooxygenase (RIFMO) catalyzes the hydroxylation of rifampicin (RIF) at the C2 atom inactivating the antibiotic activity of the molecule by preventing key contacts with the RNA polymerase target. (**C**) the monooxygenases SadA and SadC attack the sulfonamide molecules and release 4-aminophenol, which is converted by the monooxygenase SadB to 1,2,4-trihydroxybenzene that lacks antibiotic activity. (**D**) sulfonamide monooxygenase and flavin reductase are required for the cleavage of sulfamethazine producing the not active dead-end products and 4-aminophenol. (**E**) Baeyer–Villiger asymmetric oxygen insertion within the carbapenem ring of imipenem inactivates the antibiotic property of the molecule.

## Data Availability

Not applicable.

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
