# Peer review of "Monooxygenases and Antibiotic Resistance: A Focus on Carbapenems"

_biology, 2023, doi:10.3390/biology12101316_

Round 1

Reviewer 1 Report

Reviewer-report-biology-2505484

Listed below are the overview contents proposed by the authors.

Title: Monooxygenases and antibiotic resistance: a focus on carbapenems

1. Introduction

2. Antibiotic resistance mediated by bacterial enzymes

2.1. Flavin-dependent monooxygenases (FMOs)

2.1.1. Baeyer-Villiger monooxygenases (BVMOs)

2.2. FMOs and antibiotic resistance

2.2.1. Tetracyclines

2.2.2. Rifamycins

2.2.3. Sulfonamides

3. Carbapenems

3.1. Mechanism of action

3.2. Carbapenems resistance

4. BVMO and antibiotic resistance

4.1. Carbapenems

4.2 β-lactams

5. Conclusions and future directions

Major suggestions

§  This is really a small and specific topic which is a part of enzyme-mediated antibacterial resistance. So, this should be considered to move to a mini review.

§  Although, this manuscript focused on enzymes named monooxygenases. We still need to describe the molecular mechanisms that occurred in bacterial cells like genes related, -controlled, or encoded monooxygenases.

§  The topic number 4 (4. BVMO and antibiotic resistance) should be changed to be “4. BVMO and β-lactams resistance” or “4. BVMO and carbapenems resistance”. Moreover, this is a major topic as considered the title of this manuscript, authors should summarize in the table to describe the main and most common mechanisms together with representative bacterial species (especially as mentioned in Abstract: Klebsiella pneumonia, Escherichia coli, Pseudomonas aeruginosa and Acinetobacter baumannii) that reported by previous experiment-based papers.

§  Figure 3 is a low resolution image and need to be improved for readers.

no comments

Author Response

Major suggestions

  • This is really a small and specific topic which is a part of enzyme-mediated antibacterial resistance. So, this should be considered to move to a mini review.

Thank you for your suggestion regarding the scope and format of our manuscript. We appreciate your perspective on the topic's size and specificity, and we take your recommendation to consider a mini-review into careful consideration. Upon reflection, we believe that the idea of converting our work into a mini-review is indeed an interesting and valid option, given the focused nature of our research. A mini-review format would allow us to present a concise yet comprehensive overview of the specific aspect of enzyme-mediated antibacterial resistance we have investigated. This format can be beneficial in providing a condensed, reader-friendly synthesis of the existing literature, highlighting key findings, and offering valuable insights into the subject matter.

  • Although, this manuscript focused on enzymes named monooxygenases. We still need to describe the molecular mechanisms that occurred in bacterial cells like genes related, -controlled, or encoded monooxygenases.

Thank you for your additional feedback and insightful comment regarding the need to describe the molecular mechanisms in bacterial cells, including genes related to monooxygenases. We appreciate your attention to the broader context and the desire for more comprehensive coverage of this topic.

Your suggestion aligns with the importance of providing a well-rounded understanding of the subject matter. To address this aspect, we think that that a separate mini-review dedicated to the molecular mechanisms involved in bacterial cells, including the genes associated with monooxygenases, would be a valuable addition to the literature. This would allow for a more detailed exploration of the intricate processes at the molecular level, enhancing the overall comprehension of the topic.

In response to your suggestion, we will consider developing a separate mini-review that focuses specifically on the molecular mechanisms, genetic regulation, and relevant pathways related to monooxygenases in bacterial cells. This would provide a comprehensive overview of this critical aspect of enzyme-mediated antibacterial resistance.

  • The topic number 4 (4. BVMO and antibiotic resistance) should be changed to be “4. BVMO and β-lactams resistance” or “4. BVMO and carbapenems resistance”. Moreover, this is a major topic as considered the title of this manuscript, authors should summarize in the table to describe the main and most common mechanisms together with representative bacterial species (especially as mentioned in Abstract: Klebsiella pneumonia, Escherichia coli, Pseudomonas aeruginosa and Acinetobacter baumannii) that reported by previous experiment-based papers.

The section 2: Antibiotic resistance mechanism has been added.

  • Figure 3 is a low resolution image and need to be improved for readers.

Figure 3 has been improved.

Reviewer 2 Report

The submitted manuscript provides for the review of monooxygenases and their roles in antibiotic resistance, with a specific focus made with regards to carbapenem resistance. The review is well-written and very informative with regards to the subject matter. The authors have adequately summarized the clinical impact of antibiotic resistance, the various mechanisms of antibiotic resistance and have given a brief background to carbapenem resistance. There are, however, a few corrections/amendments the authors should make before this review can be suitable for publication. A list of comments/corrections has been provided below.

1.      There are a few spelling and grammar errors in the manuscript. A proof-read of the complete manuscript is advised.

2.      Figure 3: the text in this figure is not legible due to small text size. The authors are requested to improve the quality of this image.

3.      Section 4.1 and 4.2: this section is the core focus of this review and further details/discussion with regards to the studies reviewed should be provided. Additionally, the authors should provide a background regarding the incidence of carbapenem resistance and any clinical data that is available on this.

There are a few spelling and grammar errors in the manuscript. A proof-read of the complete manuscript is advised.

Author Response

The submitted manuscript provides for the review of monooxygenases and their roles in antibiotic resistance, with a specific focus made with regards to carbapenem resistance. The review is well-written and very informative with regards to the subject matter. The authors have adequately summarized the clinical impact of antibiotic resistance, the various mechanisms of antibiotic resistance and have given a brief background to carbapenem resistance. There are, however, a few corrections/amendments the authors should make before this review can be suitable for publication. A list of comments/corrections has been provided below.

  1. There are a few spelling and grammar errors in the manuscript. A proof-read of the complete manuscript is advised.

A proof read has been done.

  1. Figure 3: the text in this figure is not legible due to small text size. The authors are requested to improve the quality of this image.

Quality has been improved.

  1. Section 4.1 and 4.2: this section is the core focus of this review and further details/discussion with regards to the studies reviewed should be provided. Additionally, the authors should provide a background regarding the incidence of carbapenem resistance and any clinical data that is available on this.

The section 3.2 Carbapenem resistance in ESKAPE pathogens has been added-

Reviewer 3 Report

The manuscript is valuable and deserves publication.

Looking at the reference list, for a review, it is a short list.

The introduction is supported on only 3 refs. Some more can be added.

Antioxidants play an important role. A connection should be added.

Excepting the first 4 references, the supporting literature is not updated. A review should cover more from the recent literature.

MDPI published articles has been not consulted, even if the authors submitted their work to a MDPI journal. There are several studies discussing connected topics.

Author Response

The manuscript is valuable and deserves publication.

Looking at the reference list, for a review, it is a short list.

Thank you for your observation regarding the length of the reference list in our review. We appreciate your feedback and recognize the importance of having a robust and comprehensive set of references to support the content and context of our work. The literature section was expanded.

The introduction is supported on only 3 refs. Some more can be added.

5 references have been added.

Antioxidants play an important role. A connection should be added.

A connection has been added in the Introduction section of the manuscript.

Excepting the first 4 references, the supporting literature is not updated. A review should cover more from the recent literature.

Literature has been updated with more recent publications from the literature.

MDPI published articles has been not consulted, even if the authors submitted their work to a MDPI journal. There are several studies discussing connected topics.

6 MDPI published articles has been added.